# A transposable element insertion is associated with an alternative life history strategy

Alyssa Woronik[1,7]*, Kalle Tunström [1], Michael W. Perry [2,8], Ramprasad Neethiraj[1], Constanti Stefanescu[3,4], Maria de la Paz Celorio-Mancera [1], Oskar Brattström [5], Jason Hill[1,9], Philipp Lehmann [1], Reijo Käkelä[6] & Christopher W. Wheat[1]*

Tradeoffs affect resource allocation during development and result in fitness consequences that drive the evolution of life history strategies. Yet despite their importance, we know little about the mechanisms underlying life history tradeoffs. Many species of *Colias* butterflies exhibit an alternative life history strategy (ALHS) where females divert resources from wing pigment synthesis to reproductive and somatic development. Due to this reallocation, a wing color polymorphism is associated with the ALHS: either yellow/orange or white. Here we map the locus associated with this ALHS in *Colias crocea* to a transposable element insertion located downstream of the *Colias* homolog of *BarH-1*, a homeobox transcription factor. Using CRISPR/Cas9 gene editing, antibody staining, and electron microscopy we find white-specific expression of *BarH-1* suppresses the formation of pigment granules in wing scales and gives rise to white wing color. Lipid and transcriptome analyses reveal physiological differences associated with the ALHS. Together, these findings characterize a mechanism for a female-limited ALHS.

[1] Department of Zoology, Stockholm University, S106 91 Stockholm, Sweden. [2] Department of Biology, New York University, New York, NY 10003, USA. [3] Museum of Natural Sciences of Granollers, Granollers, Catalonia 08402, Spain. [4] CREAF, Cerdanyola del Valles, Catalonia 08193, Spain. [5] Department of Zoology, University of Cambridge, Cambridge CB23EJ, UK. [6] Helsinki University Lipidomics Unit (HiLIPID), Helsinki Institute for Life Science (HiLIFE) and Molecular and Integrative Biosciences Research Programme, University of Helsinki, FI00014 Helsinki, Finland. [7] Present address: Department of Biology, New York University, New York, NY 10003, USA. [8] Present address: Division of Biological Sciences, University of California San Diego, La Jolla, CA 92093, USA. [9] Present address: Department of Medical Biochemistry and Microbiology, Uppsala University, Uppsala, Sweden. *email: wa23@nyu.edu; chris. wheat@zoologi.su.se

A life history strategy is a complex pattern of co-evolved life history traits (e.g. number of offspring, size of offspring, and lifespan[1]), that is fundamentally shaped by tradeoffs that arise because all fitness components cannot simultaneously be maximized. Therefore, finite resources are competitively allocated to one life history trait versus another within a single individual, and selection acts on these allocation patterns to optimize fitness[2]. Evolutionary theory predicts that positive selection will remove variation from natural populations, as genotypes with the highest fitness go to fixation[3]. However, across diverse taxa alternative life history strategies (ALHSs) are maintained within populations at intermediate frequencies due to balancing selection[4]. Life history theory was developed using methods, such as quantitative genetics, artificial selection, demography and modeling to gain significant insights into the causes and consequences of genetic and environmental variation on life history traits. Yet despite these advances, a key challenge that remains is to identify the proximate mechanisms underlying tradeoffs, especially for ecologically relevant tradeoffs that occur in natural populations[5]. Here, we characterize one such mechanism underlying an ALHS in the butterfly *Colias crocea* (Pieridae, Lepidoptera) (Geoffroy, 1785).

*Colias* butterflies (the clouded sulphurs) are common throughout the Holarctic and can be found on every continent except Australia and Antarctica[6]. In approximately a third of the nearly 90 species within the genus, females exhibit two alternative wing-color morphs: yellow or orange (depending on the species) and white[6–8] (Fig. 1a). The wing color polymorphism arises because during pupal development the white morph, also known as Alba, reallocates larval derived resources from the synthesis of energetically expensive colored pigments to reproductive and somatic development[9]. This tradeoff has been well characterized in *Colias crocea*, the Old World species that we focus upon in this work, via radio-labelled metabolite tracking in pupae[10], as well as in the New World species *Colias eurytheme* (Pieridae, Lepidoptera) (Boisduval, 1852) using ultraviolet spectrophotometry[9]. As a result of the resource reallocation, Alba females exhibit faster pupal development, a larger fat body, and significantly more mature eggs at eclosion compared to orange females[11]. However, despite these developmental advantages and the dominance of the Alba allele, the polymorphism is maintained by several abiotic and biotic factors[11–15]. For example, males preferentially mate with orange females, as wing color is an important cue for mate recognition[11,13,14]. This mating bias likely has significant fitness costs for Alba females because males transfer essential nutrients during mating, and multiply mated females have more offspring over their lifetime[16,17]. The mating bias against Alba females is strongest in populations that frequently co-occur with other white Pierid butterfly species due to interference competition[14]. Also, Alba's development rate advantage is temperature dependent, with Alba females having faster development in cold temperatures[11]. Field studies confirm Alba frequency and fitness increases in species that inhabit cold and nutrient poor habitats, where the occurrence of other white Pierid butterflies is low. While in warm environments with nutrient rich host plants and a high co-occurrence of other white species, orange females exhibit increased fitness and frequency[13–15]. Previous work has also suggested Alba females have a higher sensitivity to viral infections[10]. In all *Colias* species where it has been investigated ($n =$ 6), the switch between the Alba or the orange strategy is controlled by a single, autosomal locus[6]. This fact, along with ancestral state reconstruction[7], has led to the assumption that the Alba locus is conserved within the genus *Colias*, and potentially across the subfamily Coliadinae. Yet, despite over a century of research on various aspects of Alba biology the mechanism underlying this polymorphism remained unknown[6,8].

Here we map the locus associated with the Alba polymorphism in *Colias crocea* to a transposable element insertion downstream of the *Colias* homolog of *BarH-1*, a homeobox transcription factor. We use antibody staining to confirm Alba-specific expression of BarH-1 in the scale building cells of pupal wings and use CRISPR/Cas9 gene editing to validate *BarH-1*'s functional role in the wing-color switch. We then use scanning electron microscopy to determine that *BarH-1* expression gives rise to white wing color by reducing the number of pigment granules within Alba wing scales. We find evidence that the Alba mechanism is likely conserved across the genus *Colias*, as Alba females of the North American species *C. eurytheme* also exhibit fewer pigment granules within wing scales compared to orange females. Finally, we use lipid and transcriptome analyses to characterize physiological differences associated with the ALHS in *C. crocea* and find evidence that the fitness-related traits associated with the ALHS are also conserved between *C. crocea* and *C. eurytheme*. Together these findings characterize the mechanism underlying a female-limited ALHS.

## Results

**Mapping the Alba locus**. Using a de novo reference genome for *C. crocea* that we generated via Illumina and PacBio sequencing, and three rounds of bulk segregant analyses (BSA) using whole-genome sequencing from a female and two male informative crosses for Alba, we mapped the Alba locus to an ~3.7 Mbp region. Then, with whole-genome re-sequencing data from 15 Alba and 15 orange females from diverse population backgrounds, a SNP association study fine mapped the Alba locus to a ~430 kb contig that fell within the ~3.7 Mbp locus identified using the BSA crosses (Fig. 1b). The majority of SNPs significantly associated with Alba ($n = 70$ of 72) were within or flanking a Jockey-like transposable element (TE) (Fig. 1c). We determined that the TE insertion was unique to the Alba morph in *C. crocea* by quantifying differences in read depth between morphs within and flanking the insertion (Supplementary Fig. 1) and assembling orange and Alba haplotypes for this region (Supplementary Fig. 2). We also validated the insertion by testing that reads for each morph mapped as expected across the two haplotypes (Supplementary Fig. 2 & 3) and used PCR to validate the presence or absence, respectively, of the insertion in 25 Alba and 57 orange wild-caught females (Supplementary Fig. 4). We also found no evidence of a TE insertion in the homologous region of other Lepidopteran genomes (*Bombyx mori* & *Heliconius melpomene*) (Supplementary Fig. 1).

**Functional investigation of the Alba locus**. The Alba-specific insertion was located ~30 kb upstream of a gene encoding a DEAD-box helicase, and ~6 kb downstream of the *Colias* homolog of *BarH-1*, a homeobox transcription factor (Fig. 1c). *BarH-1* was an intriguing find as it affects color via pigment granule development within eyes of *Drosophila melanogaster*[18]. To investigate BarH-1 expression in developing *C. crocea* wings, we used in situ hybridization to localize BarH-1 proteins in wings of two day old pupae of orange and Alba females. We found the BarH-1 protein is expressed in scale building cells within the white wing regions of Alba females (Fig. 2b). We did not observe BarH-1 in scale building cells from orange areas of the wing in orange females (Fig, 2c). Interestingly, however, we found BarH-1 is expressed in scale building cells within black regions for both morphs (Fig, 2a, d). To validate the functional role of *BarH-1* in the Alba phenotype, we generated CRISPR/Cas9-mediated deletions within exons 1 and 2 using a mosaic knockout (KO) approach. *BarH-1* KO gave rise to a white/orange color mosaic on the dorsal side of the wings in females with an Alba genotype (i.e. TE insertion+) (Fig. 1d), while KO males and orange females

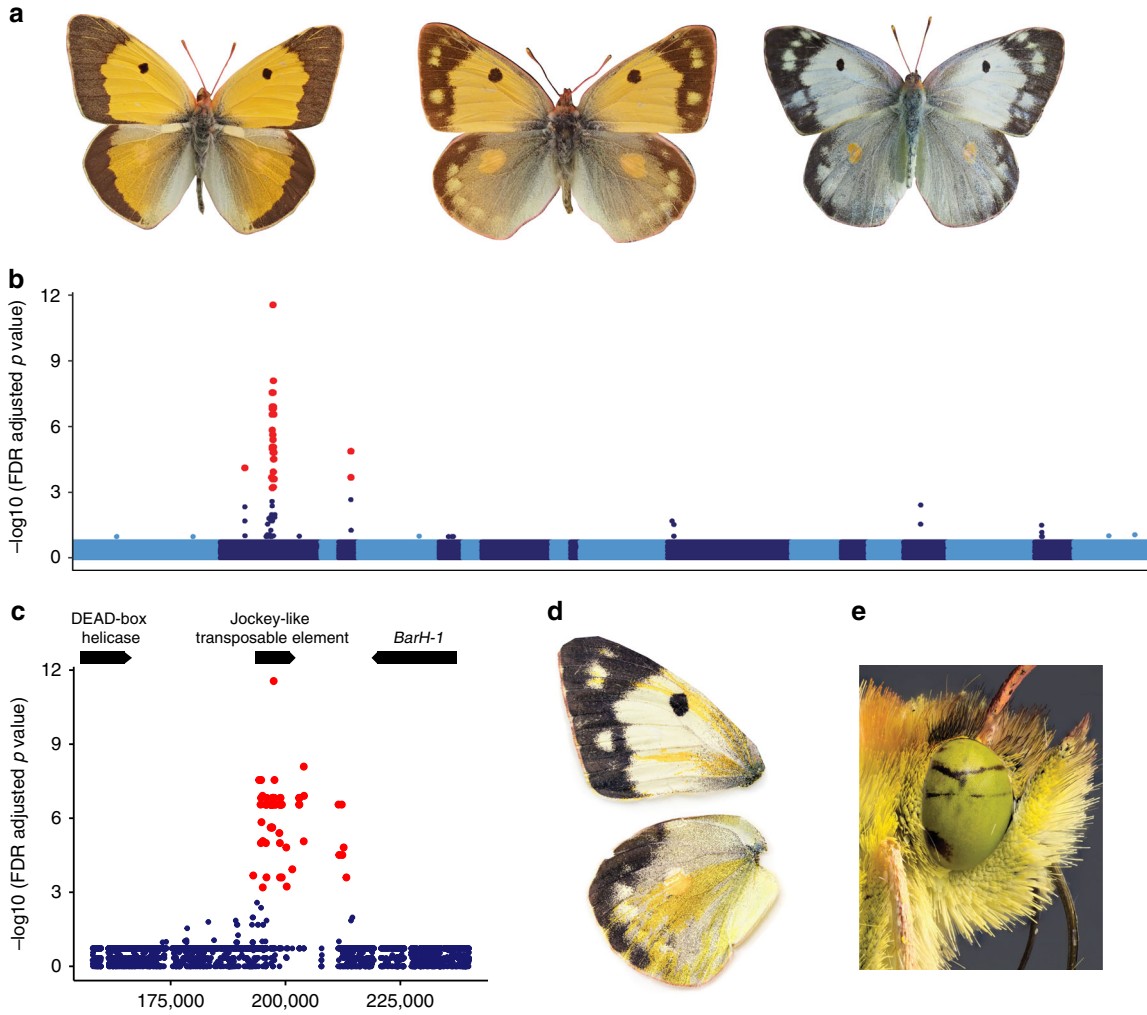

**Fig. 1 Color variation in *Colias crocea* and the genetic mechanism of Alba. a** *Colias crocea* male, orange female, and Alba female (left to right). **b** SNPs significantly associated with the Alba phenotype (red) within the ~3.7 Mbp Alba locus identified via three rounds of bulk segregant analysis. Contigs in this region shown as alternating dark and light blue. **c** The location of Alba-associated SNPs (red) on the ~430 kb outlier contig identified in the GWAS. Gene models for the DEAD-box helicase, the Jockey-like transposable element, and *BarH-1* shown at the top of the panel. **d** Wings of a female with an Alba genotype following CRISPR/Cas9 mosaic knockout of *BarH-1*, wild-type regions are white, knockout regions are orange. Orange color is seen on the dorsal forewing (top) and hindwing (bottom). **e** *BarH-1* mosaic knockout also leads to black regions in the eyes, wild-type regions are green. Source data are provided as a Source Data file. Panel **d** and **e** photo credit John Hallmén.

displayed no white/orange mosaic on the wing. These results indicate *BarH-1* expression suppresses orange coloration in the wings. We also observed black and green mosaic coloring of eyes in KO males and females of both morphs, where green eyes are the wild-type color (Fig. 1e). These results indicate *BarH-1* also plays a role in *Colias* eye development.

We next investigated how the Alba color change manifests within wings. Butterfly wing color can arise either due to the absorption of light by pigments deposited within the scales, or by the scattering of light via regularly arranged nanostructures in the scales[19]. *Colias* butterflies have pteridine pigments. These pigments are synthesized within the wings and previous work using ultraviolet spectrophotometry in *C. eurytheme* found Alba females exhibit dramatic reductions in colored pteridine pigments compared to orange[9,10,20,21]. Studies on *Drosophila* eyes indicate pteridines are synthsized in pigment granules[22,23] and pigment granules containing pteridines are concentrated within wing scales of Pierid butterflies[24]. However, whether morphs differed in wing scale morphology was unknown. To investigate wing morphology, we used scanning electron microscopy and found

white scales from Alba individuals exhibited a dramatic and significant reduction in pigment granules, compared to orange scales ($t_{5.97}$ = 2.93, $p$ = 0.03, 95% confidence interval = −25.034, 144.967, Welch two sample $t$-test, $n$ = 6 individuals, mean Alba scales = 66.9, mean orange scales = 126.9) (Fig. 3a, b). These results indicate the color change to white is caused by reduced pigment granule formation. Congruent with this interpretation, CRISPR KO Alba individuals exhibited significantly fewer pigment granules in scales from the white wild-type region compared to scales in orange *BarH-1* KO regions (Fig. 3c) ($t_{5.45}$ = 10.78, $p$ < 0.001, 95% confidence interval = 57.10, 91.70, Welch two sample $t$-test, $n$ = 10 wing scales within a single mosaic individual, mean white scales = 32.2, mean orange scales = 106.6). To further test whether reduction in pigment granule amount alone was sufficient for the orange to white color change, we chemically removed the pigment granules containing pteridines from the wing of an orange *C. crocea* female. This resulted in formerly orange regions turning white (Fig. 3d). Wings likely appear white after granule removal due to the scattering of light from the remaining non-lamellar

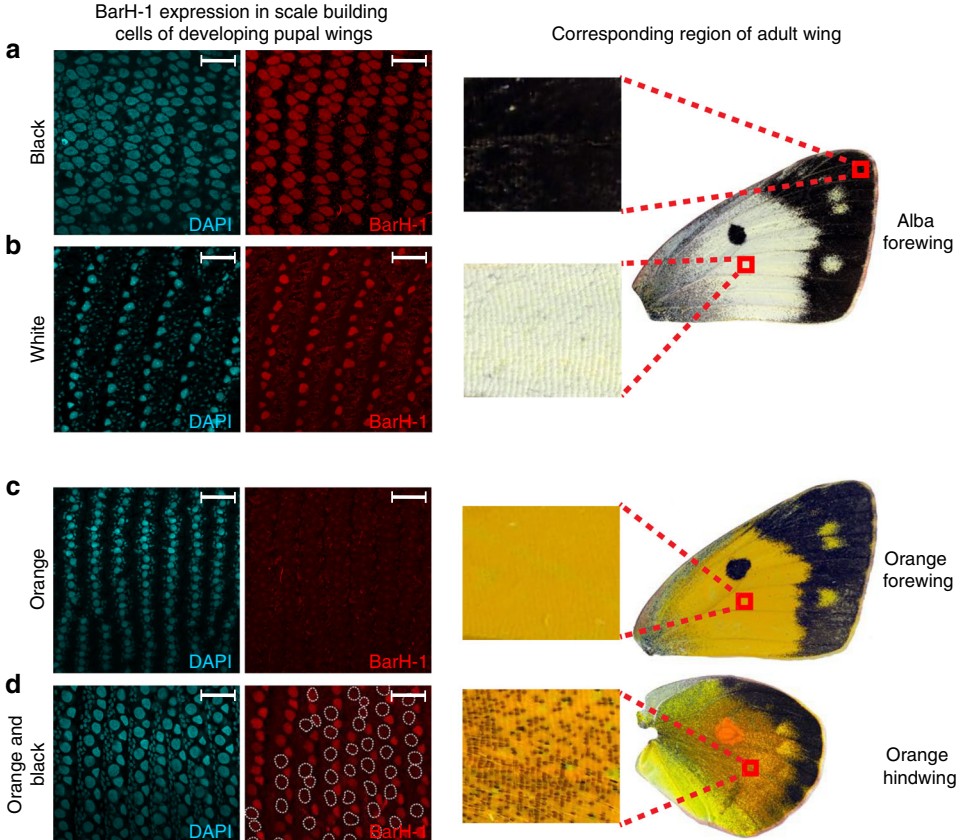

**Fig. 2 BarH-1 is expressed in white but not orange regions of the wing in *C. crocea*.** DAPI (nuclei, left, blue) and BarH-1 antibody (right, red) staining of pupal wings. Large nuclei are in scale building cells, small nuclei are in epithelial cells. The right part of the panel shows the approximate location of the stained area and the scales in this region in an adult female wing. Scale bars are 40 µM. **a** Staining of the forewing of an Alba female from black and **b** white regions. BarH-1 is expressed in black (melanic) as well as white Alba scale building cells. **c** Antibody staining of the forewing of an orange female in orange region. BarH-1 is not expressed in the orange scale building cells. **d** Antibody staining of the hindwing of an orange female. BarH-1 is heterogeneously expressed in the scale building cells within this region. This staining pattern presumably corresponds to the variation in scale color, with melanic (black) scale building cells expressing BarH-1 but orange lacking expression.

nanosctructures[25]. These results demonstrate that *BarH-1*'s supression of pigment granule formation in Alba wing scales, results in the white color of Alba females in *C. crocea*. Thus, we propose the resource tradeoff between color and development arises due to a classic Y reallocation model, wherein limited resources are competatively allocated and increased investement in one trait results in a decreased investment to another[26]. Within the energetically closed system of a developing pupa, reduced pigment granule formation would likely result in reduced pigment synthesis, which would in turn leave more resources free to be used for other developmental processes. Finally, we also observed scale building cells in black regions of both morphs express BarH-1 (Fig. 2a, d) and also lack pigment granules (Fig. 3a, b), but these scales appear black due to melanin deposition within the scale[19]. These results suggest BarH-1 may also repress pigment granule formation within black scales.

The Alba mechanism is assumed to be conserved across *Colias*. Therefore, we wished to test whether Alba females from the New World species *Colias eurytheme* also exhibited fewer pigment granules than orange females. Indeed, we found orange *C. eurytheme* scales exhibited abundant pigment granules, while Alba scales almost lacked granules (Fig. 3e, f). These results demonstrate white wing color arises via the same morphological mechanism within *Colias* and corroborate previous assumptions that Alba is conserved across the genus.

**Comparing morph physiology within and between *Colias* species.** To validate that other aspects of the Alba/orange alternative life history strategy are conserved across the genus we tested whether one of the physiological tradeoffs of Alba reported for a New World species was also seen in *C. crocea*. In *C. eurytheme*, Alba females have larger fat bodies than orange females and the strength of the Alba advantage increased in cold temperatures[11]. To compare abdominal lipid stores between morphs in *C. crocea*, we conducted high performance thin layer chromatography on 2-day-old adult females reared under two temperature treatments (Hot: 27 °C vs. Cold: 15 °C during pupal development). Adults were not allowed to feed before samples were taken, therefore these measurements reflect larval stores, where the putative energetic tradeoff should be more clearly visible. We found that recently eclosed adult Alba females had larger abdominal lipid stores than orange in both temperature treatments, though the difference was only significant in the cold treatment (cold: $n = 32$, mean Alba = 0.545. mean orange = 0.346, $t_{29.12} = 3.42$, $P = 0.002$, 95% confidence interval = 0.080, 0.318, hot: $n = 25$, mean Alba = 0.654, mean orange = 0.575, $t_{22.71} = 0.67$, $P = 0.51$, 95% confidence interval = −0.166, 0.324, Welch two sample *t*-test, two sided) (Fig. 4a). These results are consistent with previous reports from New World *Colias* species and indicate that the morph-specific tradeoff associated with the color change is also conserved across the genus.

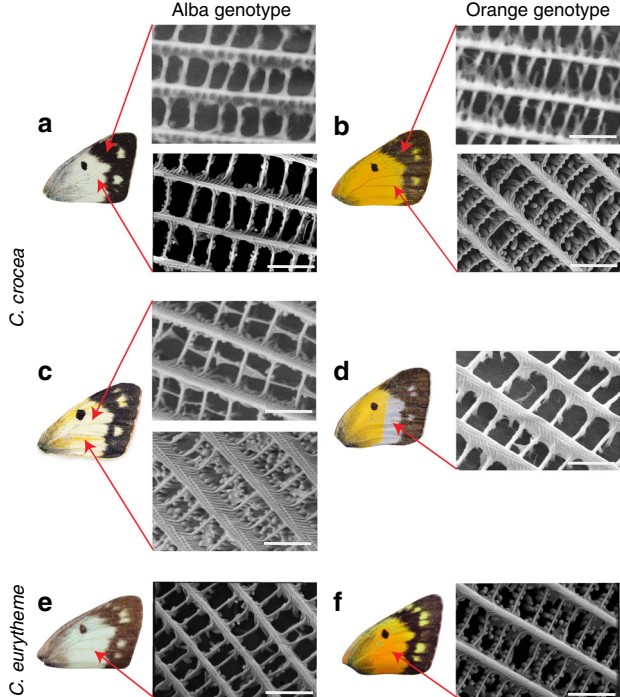

**Fig. 3 *Colias* forewings and scanning electron microscopy of wing scale nanostructures. a** *C. crocea* wild-type Alba female wing and wing scale structure. The top panel shows the scanning electron microscope (SEM) image of a black scale; pigment granules are absent. The bottom panel shows a white scale, exhibiting near absence of pigment granules. **b** Wing and wing scale structures of a wild-type orange *C. crocea* female. The top panel shows a black scale, pigment granules are absent. The bottom panel shows an orange scale with abundant pigment granules. **c** Wing and wing scales of a *C. crocea* female with an Alba genotype (i.e. transposable element insertion present) exhibiting CRISPR/Cas9 mosaic knockout of *BarH-1*. The top panel shows a wild-type white scale, where pigment granules are mostly absent. The bottom panel shows a scale in an orange *BarH-1* KO region. It exhibits significantly more pigment granules than the white scales ($t_{5.45}$ = 10.78, $p < 0.001$, Welch two sample t-test, $n =$ 10 scales in a single mosaic individual). **d** Wing and wing scale of an orange *C. crocea* female where pigment granules have been chemically removed from the distal half of the wing. The SEM image shows a scale from the white region with pigment granules completely missing. The white color of this wing section presumably results from light reflection off the remaining scale nanostructures. **e** Wing and wing scale structure of a *C. eurytheme* Alba female. Wing scales exhibit few pigment granules, similar to the phenotype observed in *C. crocea*. **f** Wing and wing scale structures of a *C. eurytheme* orange female. Orange scales show abundant pigment granules, again consistent with the orange phenotype observed in *C. crocea*. All scale bars are 2 μm.

We then investigated the transcriptome of pupal abdomen and wing tissue at the time of pteridine synthesis (>70% of pupal development, Supplementary Table 1) to identify genes that exhibited differential expression between morphs and therefore may play a role in the morph-specific differences in physiology that arise due to the resource tradeoff (Fig. 4b, c). In *C. eurytheme* Alba females emerge from the pupa with significantly more mature eggs than orange females[11] and we find evidence that suggests similar dynamics are occurring in *C. crocea*. A gene set enrichment analysis (GSEA) revealed that 'embryo development ending in birth or egg hatching' (GO:0009792, $p = 0.00072$), 'proteasome-mediated ubiquitin-dependent protein catabolic process' (GO:0043161, $p = 0.00073$), and 'proteolysis' (GO:0006508, $p = 0.00101$) were within the top 5 terms enriched and

upregulated within Alba abdomens (all p-values reported from the GSEA are the result of a Fisher's Exact Test in the R package topGO using the weight01 algorithm, Supplementary Data 1). Additionally, in our differential expression analysis a gene encoding a triacylglycerol lipase was significantly upregulated within Alba abdomen tissue (log fold change [log FC] of 4.8) (Fig. 4b). Triacylglycerol composes more than 90% of the lipids stored in the fat body and during times of energy demand triacylglycerol lipases mobilize these stores[27]. For example, during oogenesis there is a massive shift in lipid distribution from the fat body to ovaries as lipids comprise 30–40% of the dry weight of insect oocytes[27]. Taken together these results suggest that, similar to *C. eurytheme*, Alba females of *C. crocea* may be benefitting from increased oogenesis compared to orange females. We also observe an enrichment of 'defense response to Gram-positive bacterium' (GO:0050830, 0.00027) for genes upregulated within Alba abdomens. Interestingly, previous work has suggested that Alba females may have increased sensitivity to viral infection[10]. Further investigation of potential morph-specific tradeoffs between wing color and immunity is of interest.

For genes downregulated in Alba abdomens the GSEA revealed significant enrichment of 'regulation of nucleoside metabolic process' (GO:0009118, p-value < 0.0001) and 'regulation of purine nucleotide catabolic process' (GO:0033121, p-value < 0.0001) (Supplementary Data 2). *Colias* synthesize pteridines in their wings from purine precursors and Alba females exhibit dramatic reductions in colored pteridines compared to orange females[20]. Thus, downregulation of these GO terms in Alba abdomens may result from a decrease in purine precursors (guanosine triphosphate [GTP]) being shunted from the abdomen to the wings for pteridine synthesis. Additionally, consistent with previous reports of GTP reallocation from wings to other areas of development in Alba females[9] we also observed significant enrichment for 'positive regulation of GTPase activity' (GO:0043547, p-value < 0.0001). Additionally, *RIM*, a Rab3 GTPase effector[28], was one of the most highly differentially expressed (DE) genes in both tissues (logFC increase in Alba of 3.4 in the abdomen and 5.1 in the wings) (Fig. 4b, c). RIM is localized to the plasma membrane and forms a GTP-dependent complex between the membrane and vesicles to mediate calcium-regulated exocytosis[29]. If and how RIM plays a role in the Alba-associated GTP reallocation is unknown. However, RIM is known to be involved in exocytosis across diverse tissues and taxa. For example, in the neuronal synapse, RIM is involved in neurotransmitter release, a function that is evolutionarily conserved across mammals and insects[28,30]. While in the prothoracic gland of *D. melanogaster*, RIM plays an essential role in the releases the of hormone ecdysone[31]. Ecdysone is largely known for its role in regulating the timing of molting and metamorphosis in insects[32]; however, it can also affect immunity[33], longevity[34] and reproduction, specifically ovarian maturation and oogenesis[35]. The physiological changes associated with ecdysone are congruent with the fitness traits associated with Alba, thus investigating whether morphs differ in ecdysone levels may be interesting. Additionally, RIM expression has not been previously reported in wings and investigating the role it may play in this tissue is of interest.

Within wings, *BarH-1* was not differentially expressed at the time of pigment synthesis (~>70% of pupal development, Supplementary Table 1), indicating that morph-specific expression differences are temporal (antibody staining of BarH-1 in pupal wings was conducted 48 h after pupation in 27 °C with 20 h of daylight, ~30% of pupal development). However, we did observe that genes downregulated in Alba wings were significantly enriched for 'xanthine dehydrogenase activity' ($p = 0.02$, GO:0004854) (Supplementary Data 3). Xanthine dehydrogenase is the enzyme that catalyzes the xanthopterin to leucopterin

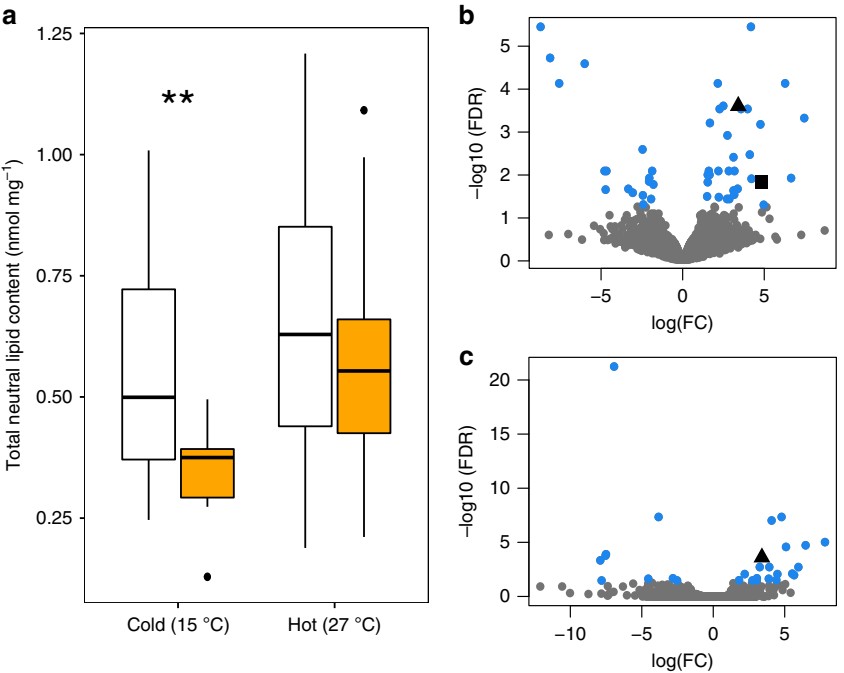

**Fig. 4 Physiological differences between female morphs of _C. crocea_. a** The mass corrected total neutral lipid content within abdomens for female morphs in two temperature treatments. White boxes are Alba females, while orange boxes are orange females. Alba females, on average, have larger neutral lipid stores in their abdomens than orange females. However there is an interaction between morph and temperature as the difference is only significant in the cold treatment (cold: $t_{29.12} = 3.42$, $P = 0.002$, $n = 32$ abdomens hot: $n = 25$ abdomens, $t_{22.71} = 0.67$, $P = 0.51$). Boxplot center line, median; box limits, 25th and 75th percentiles; whiskers, 1.5× interquartile range from the hinge; points, outliers. **b** Volcano plot to visualize gene expression differences between female morphs in pupal abdominal tissue. Each point is a gene. Genes not significantly differentially expressed between morphs are grey, while differentially expressed genes are blue. The black square is the triacylglycerol lipase and the black triangle is _RIM_. The _X_-axis is the log of the fold change (FC), positive log(FC) indicates the gene is upregulated in Alba individuals. **c** Volcano plot to visualize gene expression differences between female morphs in pupal wing tissue. Color coding, shapes, and axes are the same as in panel B. Source data are provided as a Source Data file.

conversion during pteridine synthesis in _Colias_ butterflies[9,21] and previous work in _D. melanogaster_ found xanthine dehydrogenase is localized within type II pigment granules, which synthesize and store pteridines, within the eye[22,23,36,37]. These results are consistent with previous studies in _C. eurytheme_ that reported the level of xanthopterin in Alba wings was 7–8 fold less than in orange[9]. Additionally we observed enrichment of 'MAP kinase activity' (GO:0004709, $p = 0.00109$) in genes downregulated within Alba wings. In _Drosophila_, BarH-1 represses Decapentaplegic, a morphogen that is homolog to TGFβ[38]. TGFβ can activate signalling cascades, including MAP kinase pathways[39]. Previous work in _Drosophila_ has also suggested an interaction between _Bar_ homeobox genes and Ras/MAP kinase signalling during eye development[40]. Future functional studies of the above mentioned candidate genes are needed to better understand their mechanistic roles in morph-specific development and the trade-offs associated with the ALHS.

## Discussion

Here we characterize the proximate mechanisms underlying a female-limited ALHS in a natural population. Historically, the field of life history research has treated mechanistic details as a black box[5], though recently several genetic mechanisms underlying ecologically relevant ALHSs have been identified, e.g. in the wall lizard[41], ruff[42,43], white throated sparrow[44] and fire ant[45]. The majority of these studies found that supergenes, large loci that maintain many genes in tight linkage due to structural variation, gave rise to the alternative morphs[42–46]. Such findings established that structural variation facilitates the evolution of complex traits. However these genomic architectures make

determining the specific contributions of individual genes to ALHSs difficult, though there have been significant advances made in the white throated sparrow[44]. In contrast, recent work in the wall lizard[41], found that ALHSs arose due to changes in the regulatory regions of two genes. Based on the intergenic location of the Alba-associated insertion and the Alba-specific expression of BarH-1, we predict that the insertion affects the regulation of _BarH-1_. However, this raises the question of how the locus gives rise to the other fitness-related traits associated with the ALHS. Our parsimonious hypothesis is that the Alba-associated physiological and developmental traits arise due to a classic Y reallocation model, where reduced pigment granule formation results in reduced pigment synthesis, which in turn leaves more resources free to be used for other developmental processes within the energetically closed system of the developing pupa. However, other possibilities also exisit. The other fitness-related traits may emerge due to the insertion affecting the expression of _BarH-1_ in other tissues, or surrounding genes could also have altered expression, either in the wings or other tissues. Alternatively, other mutations, located near the insertion, may cause these traits. Dissecting these details is an important avenue of ongoing research.

Previous work has shown that _BarH-1_ plays a role in the morphogenesis of neurons, leg segments, and eyes in _Drosophila_[47]. Specifically, _BarH-1_ expression is required for the formation of pigment granules and red pteridine pigments in the _Drosophila_ eye[18]. We find that _BarH-1_ also plays a role in eye and wing color in _Colias_ butterflies. However, as _BarH-1_ expression represses the formation of pigment granules within _Colias_ wings, we find it has a reversed function in _Drosophila_ and _Colias_. This may be one of several examples where either whole or a part of a

gene regulatory network that regulates eye development has been co-opted to give rise to a novel trait in the insect wing[48]. If so, future work could investigate what aspects of the network have been co-opted and how this lead to BarH-1's contrasting roles in morphogenesis.

Additionally, recent work in the field of butterfly wing evolutionary-development has found that several genes are repeatedly involved in wing color variation across distantly related species. Such genes (e.g. *optix*[49], *WntA*[50] and *cortex*[51]) form a patterning toolkit. *BarH-1* might serve as another toolkit gene for patterning wing color in butterflies beyond *Colias* as we found BarH-1 expression in scale building and socket cells of developing wings in *Vanessa cardui* pupae (Nymphalidae, Lepidoptera) (Linnaeus, 1758) (Supplementary Fig. 5). However, the functional role of BarH-1 in *V. cardui* wings remains to be determined. BarH-1 may have a novel function within *V. cardui* wings. Alternatively, the function of BarH-1 as a repressor of pigment granule formation could be conserved, as *V. cardui* scales do not have pigment granules[52]. Under the latter assumption, we would expect that BarH-1 is not expressed in closely related Pierinae species that, despite appearing white, exhibit abundant pigment granules that are primarily filled with the UV-absorbing pteridine called leucopterin[53]. Future work investigating the evolutionary history of BarH-1's co-option to the wing and function in other species could shed light on how complex traits such as ALHSs evolve.

## Methods

**Ethical approval**. This study is compliant with all relevant ethical regulations for animal testing and research. Ethical approval is not required for experiments involving Lepidoptera in Sweden, thus no approval was sought.

**Genome assembly**. An orange female and male carrying Alba (offspring from wild-caught butterflies, Catalonia, Spain) were mated in the lab. DNA from an Alba female offspring of this cross was extracted using a salting-out method[54]. DNA quality and quantity were assessed using a Nanodrop 8000 spectrophotometer (Thermo Scientific) and a Qubit 2.0 fluorometer (dsDNA BR, Invitrogen). A 180 bp insert size paired-end library (101 bp reads) was prepared (TruSeq PCR free) and sequenced on an Illumina Hiseq 4000 at the Beijing Genomics Institute (Shenzhen, China). A Nextera mate-pair library with a 3 kb insert size was prepared and sequenced on an Illumina HiSeq 2500 (125 bp reads) at the Science for Life Laboratory (Stockholm, Sweden). Raw reads were clone filtered using Stacks[55], adaptors were trimmed and low quality reads removed using the BBmap software package (Bushnell B. sourceforge.net/projects/bbmap/). Cleaned reads were used as input for the AllPaths-LG[56] assembly pipeline. High molecular weight DNA was extracted from two more Alba females from the above mentioned cross (i.e. full siblings) using a phenol chloroform based extraction protocol. Equal amounts of DNA from each individual was pooled and sent to the Science for Life Laboratory (Stockholm, Sweden) for PacBio sequencing on 24 SMRT cells (~17GB of data was produced). A Falcon[57] assembly was generated by the Science for Life Laboratory. We then used Metassembler[58] to merge our AllPathsLG and Falcon assemblies, using the AllPathsLG assembly as the primary assembly. The quality of the three genomes (AllpathsLG, Falcon, and the merged genome) were evaluated using the N50 value and the number of complete BUSCOs (Benchmarking Single Copy Orthologs) found in the assembly[59]. The merged assembly had a larger N50 and contained more complete BUSCOs than either the Allpaths or Falcon assemblies, therefore we used the merged assembly as the reference in all downstream analyses. The reference genome was annotated using MESPA[60] with the primary protein set as the input (see transcriptome assembly for description of this protein set). The contig in the merged assembly carrying the Alba locus had several N regions surrounding the locus (N's were retained from the All paths Assembly). We aligned the corresponding Falcon contig using MUSCLE[61] hosted by EMBL-EBI and hand filled N gaps based on the alignment. The same locus was identified by the GWAS both before and after gap filling.

**Bulk segregant analyses (BSA)**. *Female Informative Cross*: This data were previously published in Woronik and Wheat, 2017[62] to identify the chromosome carrying the Alba locus. Methods in brief are as follows. DNA was extracted from a wild-caught Alba mother (Catalonia, Spain) and 21 of her Alba and 21 of her orange female offspring using a salting-out method (Supplementary Methods). DNA quality and quantity for each individual was assessed via a Nanodrop 8000 spectrophotometer (Thermo Scientific, MA, USA), a Qubit 2.0 Fluorometer (dsDNA BR; Invitrogen, Carlsbad, CA, USA), and by running extracted DNA on a

1% agarose gel. Equal amounts of high-quality DNA from the 21 F1 Alba individuals was combined into a single pool, and the same was conducted for the 21 F1 orange individuals. The two morph-specific pools of DNA and DNA from the Alba mother underwent library preparation (TruSeq PCR-free) and Illumina sequencing (101 bp PE HiSeq2500), at the Beijing Genomics Institute (Shenzhen, China). Raw reads were filtered and trimmed as described in the genome assembly section. Cleaned reads were mapped to the *C. crocea* reference genome using NextGen-Map[63]. SAMTOOLS[64] was used to filter, sort, and index the bam files and generate mpileup files for the two pools and the Alba mother. Insertions and deletions were identified and masked using Popoolation2[65] and Popoolation[66], respectively. Popoolation2[65] was used to convert the F1 mpileup files to a sync files and calculate the allele frequency difference between Alba and orange pools. Resulting SNP sites from the F1 pools were filtered in R[67], for a read depth ≥20 and ≤90 and a bi-allelic state. For the Alba mother, the major and minor allele frequencies were calculated in R[67] by dividing the major and minor allele count in the mpileup file by the read depth at each site. Heterozygous SNPs in the mother were further filtered for a read depth ≥15 and ≤60. Based on the frequency of Alba to orange females in the F1 we determined the Alba mother was heterozygous for Alba (*Aa*) and had mated a male also heterozygous for Alba (*Aa*). Thus, a SNP was considered Alba-associated in the female informative cross if it met the following criteria: (1) the site was heterozygous in the Alba mother (i.e. allele frequency between 0.4 and 0.6), (2) The same SNP site was homozygous in the F1 orange pool (i.e. major allele at 100% frequency), (3) The allele frequency difference between the F1 orange pool and the F1 Alba pool was between 0.4 and 0.8, (4) The nucleotide change at the SNP site was the same in both the F1 Alba pool and the Alba mother datasets.

*Male Informative Cross I*: DNA was extracted from a wild-caught orange mother (Catalonia, Spain) and 26 of her Alba and 24 of her orange female offspring using a salting-out method (Supplementary Methods). DNA quality and quantity of each individual was assessed as in the female informative cross, before pooling equal amounts of high-quality DNA from Alba and orange offspring into two pools, respectively. Library preparation (TruSeq PCR-free) and Illumina sequencing (101 bp PE HiSeq2500), was performed on the two pools and the orange mother at the Beijing Genomics Institute (Shenzhen, China). The same read cleaning, mapping and SNP calling pipeline used on the female informative cross was applied to this dataset. Resulting SNP sites from the F1 pools were filtered in R[67], for a read depth ≥30 and ≤300, a bi-allelic state, and a minimum minor allele frequency of 3. The mother's mpileup was analyzed as described for the female informative cross and SNP sites were filtered for a read depth ≥5 and ≤30). A SNP site was considered a male informative cross I Alba SNP when it met the following expectations: (1) homozygous in the orange mother, (2) homozygous in the orange pool, (3) the allele frequency difference in the Alba pool compared to the orange was 0.45–0.55.

*Male Informative Cross II*: A male carrying Alba mated an orange female in the lab. DNA was extraced using a combination of salting-out and phenol chloroform (Supplementary Methods), then was evaluated and prepared as described above for 26 Alba and 28 orange female offspring. This resulted in two DNA pools. Library preparation (TruSeq PCR-free) and Illumina sequencing (150 bp paired-end reads with 350 bp insert, HiSeqX), was performed at Science for Life Laboratory (Stockholm, Sweden). The same read cleaning, mapping and SNP calling pipeline used on the female informative and male informative I crosses was applied, except that there was no mother sequenced for the second male informative cross. The resulting output table for the pools was filtered in R, for sites with a read depth ≥20 and ≤300, a bi-allelic state, and a minimum minor allele frequency of 3. A site was considered a male informative cross II Alba SNP if (1) the SNP site was homozygous in the orange pool, (2) the allele frequency difference in the Alba pool compared to the orange pool was 0.45–0.55.

A contig was considered Alba-associated if it had ≥3 Alba SNPs in all three crosses. Nineteen Alba-associated contigs were identified. They totaled ~3.7 Mbp and are considered the Alba BSA locus (Supplementary Table 2).

**Genome wide association study**. DNA for genome re-sequencing was extracted from 15 Alba and 15 orange females from diverse population backgrounds (Catalonia, Spain and Capri, Italy) using a salting-out protocol[54]. Library preparation was conducted using Illumina TruSeq and sequencing was conducted at the Science for Life Laboratory (Stockholm, Sweden) (150 bp paired-end reads HiSeqX). Raw reads were filtered and trimmed as described in the genome assembly section. Cleaned reads were mapped to the annotated reference genome using NextGen-Map[63]. Bam files were filtered and sorted using SAMTOOLS[64]. A VCF file was generated using SAMTOOLS[64] and bcftools[64]. Read depth per site was calculated using VCFtools[68]. VCFtools was then used to call SNP sites with no more than 50% missing data, an average read depth between 15-50 across individuals, and a minimum SNP quality of 30. An association analysis was performed with PLINK[69] and a Benjamini & Hochberg step-up FDR control was applied. SNPs with FDR < 0.05 were considered Alba SNPs. We conducted this analysis both genome wide and only within the BSA locus. Both analyses fine mapped the Alba locus to the same genomic region.

**Validating the Alba insertion**. Synteny, or gene order, is highly conserved within Lepidoptera. Thus, to validate that the contig carrying the Alba locus (*C. crocea* contig 12) was properly assembled we compared gene order across homologous

regions in *Bombyx mori* (chromosome 15) and *Heliconius melpomene* (scaffold Hmel211009) by doing a tblastn search against Kaikobase v.3.2.2, default settings) and blastp against LepBase, respectively, using protein sequences that were annotated to *C. crocea* contig 12 (Supplementary Fig 1A&B). Next, an analysis of read depth using the 15 Alba and 15 orange re-sequeucing datasets mapped to our high-quality reference genome indicated that the locus was an Alba-specific insertion (Supplementary Fig 1C). Within this predicted insertion, MESPA annotated a *Jockey-like* transposable element (TE). To validate orange females lacked a TE insertion in this region we assembled the orange haplotye by performing a de novo genome assembly on the wild-caught, orange mother of male informative cross I using CLC Genomics Workbench v.5 (kmer size = 25, bubble size = 2000, https://www.qiagenbioinformatics.com/). MESPA (version 17_Aug_15)[60] was used to annotate the resulting genome assembly using the primary protein set (see transcriptome assembly for more about this protein set). We identified the orange contig carrying the *C. crocea* BarH-1 homolog and aligned it with the Alba-associated contig from our high-quality reference genome using SLAGAN alignment via wgVISTA[70–73]. Regions of conservation between the two haplotypes were observed on both sides of the insertion, but not within, and neither MESPA nor a BLAST search could annotate a TE on the orange contig (Supplementary Fig 2). As a final bioinformatic validation we mapped the whole-genome re-sequencing data to the orange assembly using SNAP[74] (-so -t 30 -F a - = -s 100 1000) (Supplementary Fig 2B&C). Reads from all orange individuals and some of the reads from 12 of the 15 Alba individuals, could properly map across the predicted insertion site on the orange haplotype. We surmised that the reads from the Alba individuals that could map properly on the orange haplotype likely arose from an orange allele and that these individuals were heterozygous for Alba (*Aa*). To test this we looked whether these reads could map to Alba haplotype. Due to max 1000bp insert size imposed via SNAP, these reads did not properly map to the Alba haplotype, lending support to our hypothesis (Supplementary Fig 2). To further test that these individuals were heterozygous for Alba, we compared the read depth within the insertion to the read depth of a nearby conserved genomic region. We found read depth within the insertion to be about half the read depth within the conserved genomic region for these individuals, lending further support to our prediction that they were heterozygous for Alba (Supplementary Fig 3). Finally, we validated the insertion using PCR. We developed primers that spanned the insertion (i.e. one primer sat within the insertion the other right outside) (Alba F: TGTGGACGTAGGTATGAGCT, Alba R: TGTCAATTAGTCCAGCAGAAA TG). Primers amplifying a region of Cytochrome C were used as a positive control (F: GGATCACCTGATATAGCATTCCC, R: CCTGGTAAAATTAAAATATAAA CTTC). We tested the marker on 118 wild-type *Colias crocea* (36 males, 57 Orange females and 25 Alba females). The primer amplified the expected fragment size across all Alba and no orange individuals (Supplementary Fig 4).

**Antibody generation and staining**. A Rabbit-anti-BarH-1 antibody was generated against the full length sequence of the *Vanessa cardui* BarH-1 homolog. Protein was generated by GenScript (Piscataway, NJ) and purified to >80% purity. DNA sequences to produce this protein were codon-optimized for bacterial expression and made via gene synthesis. GenScript injected resultant protein into host animals, collected serum for testing, and affinity purified the product using additional target protein bound to a column. Antibody staining was as described previously for *Drosophila* and butterfly tissues[54]. In brief, staged pupal wings were dissected and fixed 48 h post-pupation. Pupal wings were fixed using 4% paraformaldehyde in 1xPBS, then rinsed twice with 1xPBST, then again after 30 min on a shaker at room temperature. For blocking, PBS was replaced with 200 uL of 5% normal serum in PBST and tissue was incubated for 20 min on a shaker at room temperature. The blocking solution was then replaced with primary antibody solution and incubated overnight at 4 °C. The primary antibody solution was then replaced with PBST, rinsed three times, and was washed on a shaker at room temperature for at least one hour with 3–4 additional solution changes. Wash solution was replaced with secondary antibody solution and incubated on a shaker either overnight at 4 °C or 2 h at room temperature. The antibody solution was replaced with PBST and rinsed three times. Washing in PBST continued for at least 1 h on a shaker at room temperature. The Rabbit-anti-Bar antibody was used at 1:100, followed by secondary antibody staining with AlexaFluor-555-anti-Rabbit secondaries (Thermo Fisher Scientific, Waltham, MA, USA, Cat# A-31572) at 1:500 and counterstaining with DAPI at 1ug/uL in 1x PBS. Images were captured using standard confocal microscopy on a Leica SP5.

**CRISPR/Cas9 knockouts**. The guide-RNA (gRNA) sequences were generated by first manually looking for PAM-sites (NGG) in the exon region of *BarH-1*. Preference was given to sites in the first half of the gene and near the ends of the exons (Supplementary Table 3). Uniqueness of the target regions was confirmed using a NCBI nucleotide blast (ver. 2.5.0 + using blastn-short flag and filtering for an e-value of 0.01) against the *C. crocea* reference genome. gRNA constructs were ordered from Integrative DNA Technologies (Coralville, Iowa, USA) as DNA (gBlocks). Full gRNA constructs had the following configuration: an M13F region, a spacer sequence, a T7-promotor sequence, the target specific sequence, a Cas9 binding sequence, and finally a P505 sequence. Upon delivery, gBlocks were amplified using PCR. For each gBlock, four 50 ul reactions were conducted using the M13f and P505 primers (P505: AAAAAAAGCACCGACTCGGTGCC, M13f:

GTAAAACGACGGCCAG) and Platinum Taq (Invitrogen cat. 10966-034). The four reactions were then combined and purified in a Qiagen Minelute spin column (cat. 28004, Venlo, Netherlands). The resulting template was transcribed using the Lucigen AmpliScribe T7-flash Transcription Kit from Epicentre/Illumina (cat. ASF3507, Madison, WI, USA) followed by purification via ammonium acetate precipitation. Products were resuspended with Qiagen buffer EB, concentrations were quantified by Qubit and further diluted to 1000 ng/µl. They were then mixed with Cas9-NLS protein (PNA Bio, Newbury Park, CA, USA) and diluted to a final concentration of 125–250 ng/µl. *C. crocea* females (*n* > 40) from Aiguamolls de l'Empordà, Spain were captured and kept in morph-specific flight cages in the lab at Stockholm University where they oviposited on alfalfa (*Medicago sativa*). Eggs were collected between 1–7 h post-laying, removed from the leaf with a paintbrush, and sterilized in 7% benzalkonium chloride for ~5 min. Eggs were then rinsed in water and dried before being attached to a glass slide. Eggs were held in place on the slide using double-sided tape. Needle tips were broken using forceps and front-loaded with 0.8–1 µl of the construct (gRNA mixed with Cas9). Injections were either at a concentration of 125 or 250 ng/ul and conducted using a M-152 Narishige micromanipulator (Narishige International Limited, London, UK) with a 50 ml glass needle syringe, with injection pressure applied by hand via a syringe fitting. For more details regarding this protocol see Perry et. al. 2016[75]. Slides with injected eggs were placed in petri dishes together with a damp paper towel and placed in a sealed plastic container with another piece of damp paper towel in order to maintain high humidity until the eggs hatched. Freshly hatched larvae were transferred to fresh *M. sativa* plants using a paintbrush and kept in a climate room at 27 °C with 20 h of daylight. Upon eclosion, adults were visually expected for knockout phenotypes. To see results from various gblocks see Supplementary Table 4.

**CRISPR/Cas9 validation**. To validate the mutation, Cas9 cut sites were PCR-amplified and a ~370 bp region centered on the intended cut site was sequenced using Illumina MiSeq 300 bp paired-end sequencing. PCR primers were designed using Primer3. DNA was isolated from KO-individuals using KingFisher Cell and Tissue DNA Kit from ThermoFisher Scientific (N11997) and the robotic Kingfisher Duo Prime purification system. DNA quality and quantity were assessed via a Nanodrop 8000 spectrophotometer (Thermo Scientific, MA, USA) and a Qubit 2.0 Fluorometer (dsDNA BR; Invitrogen, Carlsbad, CA, USA). Aliquots were then taken and diluted to 1 ng/ul before amplifying the region over the cleavage-site. Sequences were amplified and ligated with Illumina adapter and indexes in a two-step process following the protocol provided by Science for Life Laboratories (Stockholm, Sweden) and Illumina. First, we amplified the ~370 bp long sequence around the cut sites and attached the first Illumina adapter, onto which we later attached Illumina handles and index using a second round of PCR (Accustart II PCR Supermix from Quanta Bio [Beverly, MA, USA], settings 94 C x 2 min followed by 40 cycles of 94 C x 30 sec + 60 C x 15 sec + 68 C x 1 min followed by 68 C x 5 min). PCR products were purified using Qiagen Qiaquick (Cat. 28104). Concentration and quality of the product were assessed via Nanodrop and gel electrophoresis. DNA was diluted to ~0.5 ng/ul and then the unique double indices were attached by the second round of PCR (same protocol as above). The final PCR products were purified again using Qiaquick spin columns and concentration and size was assessed using Qubit fluorometer and gel electrophoresis. All samples were then mixed at equal molarity and sent for sequencing at Science for Life Laboratories (Stockholm, Sweden). Sequences were aligned to their respective fragments (area surrounding cut site) using SNAP (ver. 1.0beta18)[74], identical reads were clustered using the collapser utility in Fastx-Toolkit. Sequences containing deletions were extracted and the most abundant sequences containing deletions were selected for confirmation of deletion in the expected region (Supplementary Fig. 6).

**Pigment granule removal from wings**. This protocol was published in Rutowski et. al. 2005[25]. In brief, to remove the pterins, and consequently the pigment granules, from the wings of orange *Colias* females, wings were removed from dead females and throughly wetted with 70% isopropyl alcohol. The wing was then dipped in 1% NH₄OH solution made with reagent grade water. Wings were then placed on a paper towel under a glass slide to dry.

**Electron microscopy**. To quantify pigment granule differences between wild-type Alba and orange individuals pieces of the forewing were mounted on aluminum pin stubs (6 mm length) with the dorsal side upwards. Samples were coated in gold for 80 sec using an Agar sputter coater and imaged under 5 kV acceleration voltage, high vacuum, and ETD detection using a scanning electron microscope (Quanta Feg 650, FEI, Hillsboro, Oregon, USA). The same protocol was used for the wing where pigment granules were chemically removed. To quantify pigment granules for each individual (three Alba and three orange) we selected images from the same magnification and randomly placed three 4 µm² squares on the scale image. We counted the number of pigment granules within each square and took the average (Supplementary Table 5), then conducted a Welch two sample *t*-test in R. For additional SEM images that document the variation in pigment granule density within Alba scales see Supplementary Fig. 7. To quantify pigment granule differences between KO and wild-type regions in our CRISPR KO mosaic individual, a biopsy hole punch 2 mm in diameter was used to cut out one piece mostly

containing white scales and one piece with mostly orange scales. These pieces were first photographed using a Leica EZ4HD stereo microscope in order to allow us to confirm the color of each scale once they were covered with gold sputter (Supplmentary Fig. 8). Five white and five orange scales were then selected and the granules within a 4 μm² square placed over the tip of the left lobe for each scale were counted (Supplementary Table 6). A Welch two sample *t*-test was then conducted in R.

**Lipid analysis**. Wild caught *C. crocea* Alba females (Catalonia, Spain) oviposited in the lab on *M. sativa*. Eggs were moved into individual rearing cups and randomly split between two temperature treatments (hot: 27 °C and 16 h day length during larval and pupal development, cold: reared at 22 °C with a 16 h day length during larval development and 15 °C with a 16 h day length during pupal development). Once pupated, individuals were checked a minimum of every 12 h. Upon eclosion adults were stored at 4 °C until the next day to provide time for meconium excretion. Butterflies were not allowed to feed before dissection. Body weight was taken using a Sauter RE1614 scale before dissection. Total lipids were extracted using the Folch method[76]. In brief, frozen butterfly abdomens were first homogenized in 1 ml aliquot of chloroform:methanol:water (2:1:0.2, the tissue water estimate included) in Eppendorf tubes using a TissueLyser Bead Homogenizer (Quiagen, Hilden, Germany), and then the lysate was transferred to 10 ml glass tubes and 4.5 ml of chloroform:methanol (2:1) was added. The chloroform and aqueous phases separated when 0.9 ml water was added, and after mixing and centrifugation steps the lower chloroform phase with the lipids was recovered. The remaining aqueous phase was re-extracted with the theoretical lower phase, and the recovered chloroform phases from the two extraction steps were combined, evaporated under nitrogen and dissolved into 750 μl of chloroform:methanol (1:2). This sample solution (stored in deactivated glass vials for a maximum of three days at −20 °C) was used for high performance thin layer chromatography (HPTLC) analysis. The HPTLC was conducted as follows: 5 μl of the sample lipid extract was applied on a silica plate with a Camag Automatic TLC Sampler 4 (Camag, Muttenz, Switzerland). After the silica plate developed it was scanned with a Camag TLC plate scanner 3 at 254 nm using a deuterium lamp with a slit dimension of 6 × 0.45 mm and analyzed with the Win-CATS 1.1.3.0 software. Peaks representing the four major neutral lipid classes (diacylglycerols, triacylglycerols, cholesterol and cholesterol esters) were identified by comparing their retention times against known standards. Then the peak areas were integrated and the amount of lipid within each class was calculated using the formula: $pmol_{sample} = (Area_{sample}/Area_{standard}) \times pmol_{standard}$. The total lipid content (nmol per abdomen) was calculated as a sum of pmol contents of all neutral lipid classes. For the statistical analyses this value was regressed against abdomen weight and standardized residuals (i.e. mass-corrected storage lipid amount). For more details regarding methods see Woronik et. al. 2018[12].

**Transcriptome assembly, differential expression, and gene set enrichment analysis**. Offspring from a wild-caught Alba female from Catalonia, Spain were reared at Stockholm University. When larvae reached the fifth instar they were checked at least every 6 h and the pupation time of each individual was recorded. Tissue was collected between 82% and 92% of pupal development (Supplementary Table 7). Pupae were dissected in 1xPBS solution, and the abdomen and wings were flash frozen in liquid nitrogen and stored at −80 ºC. RNA was extracted from the abdomen and wing tissues using Trizol. RNA quality and quantity was assessed using a Nanodrop 8000 spectrophotometer (Thermo Scientific) and an Experion electrophoresis machine using the manufacturer protocol (Bio-Rad, Hercules, CA). Library preparation (Strand-specific TruSeq RNA libraries using poly-A selection) and sequencing (101 bp PE HiSeq2500—high output mode) was performed at the Science for Life Laboratories (Stockholm, Sweden). In total 16 libraries were sequenced (4 orange and 4 Alba individuals—wings and abdomen from each individual). Raw reads were adaptor filtered and trimmed using the BBmap software package (Bushnell B. sourceforge.net/projects/bbmap/). Cleaned reads from all libraries were used for a de novo transcriptome assembly using Trinity[77]. To reduce the redundancy among contigs and produce a biologically valid transcript set, the tr2aacds pipeline from the EvidentialGene software package[78] was run on the raw Trinity assembly. The tr2aacds pipeline utilizes five steps to produce an optimal set of transcripts based on their coding potential. First it predicts the coding DNA sequence and the amino acid sequence of each transcript, second it removes full length redundant contigs, third it uses substring de-replication to remove redundant fragments, fourth it clusters highly similar sequences into loci and finally classifies sequences as 'okay primary', 'okay alternate', or 'drop' sequences. The 'okay primary' sequence set was used as the refernce transcriptome in all downstream analysis and called the primary set. The sixteen RNA-Seq libraries were mapped to the reference transcriptome using NextGenMap[63]. SAMTOOLS[64] was then used to filter, sort and index the sixteen bam files. SAMTOOLS[64] idxstats was then used to calculate the read counts per gene for each of the sorted bam files. These counts were then joined in a CSV file using csvjoin. A differential expression analysis was conducted in R using EdgeR[79]. A Benjamini Hochberg correction was applied to the raw p-values to correct for false discovery rate and differentially expressed genes were called (adjusted *p*-value < 0.05) (see Source Data). eggNOG-mapper[80] was used with default settings to functionally annotate the transcriptome (Supplementary Data 5). The R package topGo[81] was

used to conduct a gene set enrichment analysis on genes that exhibited >1 or <−1 log fold change in the differential expression analysis (Supplementary Data 1-4). For R code see Supplementary Methods.

**Reporting summary**. Further information on research design is available in the Nature Research Reporting Summary linked to this article.

## Data availability
Raw reads, the reference genome and transcriptome can be accessed at NCBI Genbank nucleotide database using the following accession codes. Reference genome asembly: PRJNA588020, Female informative cross: PRJNA587696, Male informative cross I: PRJNA587518, Male informative cross II: PRJNA587716, 30 resequenced Alba and orange individuals: PRJNA587791, Transcriptome and RNA-Seq: PRJNA587755. Full SEM imgaes are available on FigShare https://doi.org/10.6084/m9.figshare.10255664.v1.

## Code availability
Software versions and parameters can be found in the Supplementary Methods.

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

## Acknowledgements

We thank Karin Kiontke, Christen Bossu, Naomi Keehnen, and Peter Pruisscher for helpful comments on the manuscript. We thank Lovisa Wennerström, Elishia Harji, Jofre Carnicer, and Christina Hansen Wheat for help with fieldwork. We thank Marianne Ahlbom for assistance with the SEM. We thank the Department of Zoology at Stockholm University, the Swedish Research Council 2012–3715, the Academy of Finland 131155, the Knut and Alice Wallenberg Foundation 2012.0058 and the Erik Philip-Sörensens foundation for funding. Sequencing was performed at SciLifeLab, National Genomics Infrastructure, Stockholm and Uppsala, Sweden.

## Author contributions

A.W. conducted butterfly rearings and lab work, analysed the data, and wrote the manuscript with C.W.W. and input from the coauthors. A.W., M.W.P., K.T., and C.W.W. conducted the CRISPR/Cas9 knockout experiment. A.W. and K.T. conducted the electron microscopy. M.W.P. conducted antibody staining. R.N. and J.H. assisted with bioinformatics. P.L. and R.K. conducted HPTLC and A.W. and P.L. analyzed the data. A.W., C.S., C.W.W. and O.B. conducted fieldwork. M.C. conducted lab work. C.W.W. supervised the work at all stages.

## Competing interests

The authors declare no competing interests.
