## [Peer Review File · Nature Communications]

Reviewers' Comments:

Reviewer #1:

Remarks to the Author:

This solid and well-written manuscript contains an in-depth investigation of the genetic, molecular, developmental, and morphological basis of an alternative life history strategy (ALHS) displayed by *Colias crocea* butterflies, that seems to also apply to other species in the genus. This ALHS produces butterflies of two different colors, yellow (or orange) and white, who also display a variety of other associated life-history differences.

The biology behind this ALHS has been studied for over 100 years but this is the first time the genetic basis of the wing color polymorphism has been identified. The authors perform crosses and sequence the genomes of the two butterfly forms using a variety of lab bred and field caught specimens. They identify a transposable element insertion next to the BarH-1 transcription factor that is present in white forms but absent in yellow forms of the butterfly. Associated with the insertion is the activation of BarH-1 in white scale cells (one of the morphs of the butterfly), and its absence in yellow scale cells (the other morph). CRISPR-Cas9 targeting BarH-1 shows that this gene is sufficient to modify the appearance of white butterflies into yellow butterflies. This gene represses pteridine pigment granule development in yellow scales, and changes the color of the butterfly from yellow to white.

As a result of this genetic alteration, the authors argue that a suite of downstream effects also take place, primarily driven by resources no longer used for pigment granule production being shifted to fat body production and egg development. This is investigated at the molecular level by examination of lipid and transcript changes in abdomens and wings of wildtype white and yellow forms, at the time that pteridine pigments are being formed. Note that all these changes are inferred to be caused by the single TE insertion, but the lipidome and transcriptome of a TE crispant (where the TE present in the genome of the white form is actually removed) and the lipidome and transcriptome of a BarH-1 crispant, are not actually examined and compared with those of the two wild-type butterfly forms. It remains possible that the TE insertion disrupts the function of more than one of the genes flanking it (in wings or other body parts), and that additional mutations, in close proximity to this TE insertion, might also contribute to the phenotypes examined.

Regardless of the shortcomings above, which should probably be stated in the discussion of the manuscript at some stage, this is a solid and important contribution to the field of alternative life history strategies, which has, for the most part not been focusing on the genetic basis of the tradeoffs that are central to the field. This is also a nice addition to the field of butterfly wing color pattern evolution and development.

Minor edits

Line 31: remove one "specific".

Line 105: Did you localize mRNA (via in situ hybridization) or proteins via antibody stainings? I believe you did the latter.

Line 132: "fewer" pigment granules (as they were actually counted) than "less". This use of "less" happens again a few times further down.

Line 172: Specify what time of development this is.

Line 182: Sentence needs attention: "Identified a that gene"

Line 186: Do you mean during "oogenesis" instead of "embryogenesis" and insect "eggs" instead of "embryos"? The latter take place outside the female's body.

Line 189 – "embryogenesis"?...same comment as above.

Line 200: "the decreased pteridine synthesis observed in Alba females, which likely arises due to the decrease in pigment granules in the wings". Isn't it more logical that the decrease in pteridine synthesis causes the decrease in pigment granules?

Line 210: Some further explanation about the role of synaptic vesicle exocytosis and secretory pathways in relation to GTP reallocation are needed here.

Line 212: remind the reader about the stage being examined here and the stage where BarH-1 was first examined to be differentially expressed (don't make readers go all the way to the methods section).

Reviewer #2:

Remarks to the Author:

This article presents original findings on the mechanisms underlying a life-history and pigmentation traits in sulphur butterflies. The trait segregates naturally in the wild and causes an absence of pterin granules in the wing, which as discussed seem to benefit the reproductive of these females with increased mass and egg yield among other traits. This paper identifies BarH-1 as the clear causal gene for this polymorphism, using a combination of association mapping, immunostainings, and CRISPR mosaic KO. All the evidence is logically presented and well supported: in particular, the protein expression in the presumptive scales of the wing tissue, and the pleiotropic effect of the knock-outs on both eyes (a function conserved with *Drosophila*) and in the wings.

BarH-1 appear to have evolved as a repressor of pterin granule formation in the wings, and a transposon may drive this gain of function, which is very interesting.

Overall I find the article to be a great evolutionary biology story that combines explanation about the why and how of the Alba trait. I think this is exactly the kind of discovery that could go into evolutionary biology textbooks: not only it put in the spotlight an example of a pleiotropic trait, possibly under balance selection to maintain its polymorphism, it is also a neat example of a transposon-mediated novelty.

I especially like the key experimental workflow that goes from phenotypes to genotypes (forward genetics): from mapping to a possible gene, to a characterization of that gene. It is similar to another recent article (Westerman *Current Biology* 2018) which identified another transcription factor switching yellow/white color in completely different butterflies bearing another kind of pigments. The two stories will be interesting to compare in the classroom for student interested in evolutionary genetics and functional genomics, so I plan to personally use these two stories for pedagogic goals.

It will be interesting to determine if the transposon is found in other species thought to display this ancient Alba polymorphism, and also, perhaps to knock-out the transposon itself, but this is well beyond the scope of the present study, which I recommend for publication.

Stockholm
University

November 6, 2019

Please find below a point-by-point response to the referee's comments.

First, we would like to start by thanking the referees for taking the time to review our manuscript. We are very happy that they feel that the manuscript is an important contribution to the field and we thank them for their comments, which we feel have helped us to improve the quality of this work. The reviewer's comments are bolded and our responses directly below.

Reviewer #1 (Remarks to the Author):

This solid and well-written manuscript contains an in-depth investigation of the genetic, molecular, developmental, and morphological basis of an alternative life history strategy (ALHS) displayed by *Colias crocea* butterflies, that seems to also apply to other species in the genus. This ALHS produces butterflies of two different colors, yellow (or orange) and white, who also display a variety of other associated life-history differences.

The biology behind this ALHS has been studied for over 100 years but this is the first time the genetic basis of the wing color polymorphism has been identified. The authors perform crosses and sequence the genomes of the two butterfly forms using a variety of lab bred and field caught specimens. They identify a transposable element insertion next to the *BarH-1* transcription factor that is present in white forms but absent in yellow forms of the butterfly. Associated with the insertion is the activation of *BarH-1* in white scale cells (one of the morphs of the butterfly), and its absence in yellow scale cells (the other morph). CRISPR-Cas9 targeting *BarH-1* shows that this gene is sufficient to modify the appearance of white butterflies into yellow butterflies. This gene represses pteridine pigment granule development in yellow scales, and changes the color of the butterfly from yellow to white.

As a result of this genetic alteration, the authors argue that a suite of downstream effects also take place, primarily driven by resources no longer used for pigment granule production being shifted to fat body production and egg development. This is investigated at the molecular level by examination of lipid and transcript changes in abdomens and wings of wildtype white and yellow forms, at the time that pteridine pigments are being formed. Note that all these changes are inferred to be caused by the single TE insertion, but the lipidome and transcriptome of a TE crispant (where the TE

present in the genome of the white form is actually removed) and the lipidome and transcriptome of a *BarH-1* crispant, are not actually examined and compared with those of the two wild-type butterfly forms. It remains possible that the TE insertion disrupts the function of more than one of the genes flanking it (in wings or other body parts), and that additional mutations, in close proximity to this TE insertion, might also contribute to the phenotypes examined.

Regardless of the shortcomings above, which should probably be stated in the discussion of the manuscript at some stage, this is a solid and important contribution to the field of alternative life history strategies, which has, for the most part not been focusing on the genetic basis of the tradeoffs that are central to the field. This is also a nice addition to the field of butterfly wing color pattern evolution and development.

We thank the reviewer for their kind words and for taking the time to review our manuscript. We are glad that they feel it makes an important contribution to the field. The issues raised regarding the other possible pleiotropic effects of the TE insertion are indeed very important. Thus, we have added several lines to the discussion addressing this issue, specifically mentioning the alternatives suggested by the reviewer (lines 284-287).

Line 31: remove one “specific”.

We have removed one specific (line 35).

Line 105: Did you localize mRNA (via in situ hybridization) or proteins via antibody stainings? I believe you did the latter.

We localized proteins and have added this clarification to the text (line 127).

Line 132: “fewer” pigment granules (as they were actually counted) than “less”. This use of “less” happens again a few times further down.

We have changed the word “less” to “fewer” in all instances (lines 97, 155, 175).

Line 172: Specify what time of development this is.

We have edited this line to specify that samples were taken after 70% of pupal development. Our preliminary data, which resulted from visually inspecting the wing pads of pupa every six hours during development, indicates that color deposition in the wings begins on average at 70% of pupal development (line 198).

Line 182: Sentence needs attention: “Identified a that gene”

We have edited the sentence (line 208-209).

Line 186: Do you mean during “oogenesis” instead of “embryogenesis” and insect “eggs” instead of “embryos”? The latter take place outside the female’s body.

Line 189 – “embryogenesis”?...same comment as above.

Yes, thank you for the comment. The words oogenesis and eggs are the correct terms here. We have changed this text (lines 212, 214, and 215).

Line 200: “the decreased pteridine synthesis observed in Alba females, which likely arises due to the decrease in pigment granules in the wings”. Isn’t it more logical that the decrease in pteridine synthesis causes the decrease in pigment granules?

There is evidence from the *Drosophila* literature that the granules themselves are not only the sites where pteridines are stored, but are also where they are synthesized (References 22, 23, and 37 in the main text), hence why we hypothesized that the decrease in granules leads to the decrease in pteridine synthesis. However the reviewer is correct, we cannot identify specific causation direction for the lack of pteridines and granules and have edited the text accordingly (lines 225-229).

Line 210: Some further explanation about the role of synaptic vesicle exocytosis and secretory pathways in relation to GTP reallocation are needed here.

We have added several lines to the text that give more detail regarding what is known regarding the function of RIM. RIM plays a conserved role in synaptic vesicle exocytosis in the nervous system from mammals to *Drosophila* (References 28, 29, 30 in the Main Text). Additionally, recent work found it is also involved in exocytosis of the hormone ecdysone from the prothoracic gland in *Drosophila* (Reference 31 in the Main Text). We do not want to take too many liberties hypothesizing the function of RIM, aside from exocytosis, as the increase in RIM expression may be increasing Rab3 (its associated RabGTPase) activity, which could be driving increased GTP usage. Alternatively, whatever exocytotic process RIM is involved in could be reallocating resources. However, we do note that many of the fitness traits associated with Alba are congruent with physiological changes known to be associated with ecdysone. (lines 233-246)

Line 212: remind the reader about the stage being examined here and the stage where BarH-1 was first examined to be differentially expressed (don’t make readers go all the way to the methods section).

We have edited the text so the developmental time is mentioned in this sentence (line 250-251).

Reviewer #2 (Remarks to the Author):

This article presents original findings on the mechanisms underlying a life-history and pigmentation traits in sulphur butterflies. The traits segregates naturally in the wild and causes an absence of pterin granules in the wing, which as discussed seem to benefit the reproductive of these females with increased mass and egg yield among other traits. This paper identifies BarH-1 as the clear causal gene for this polymorphisms, using a combination of association mapping, immunostainings, and CRISPR mosaic KO. All the evidence is logically presented and well supported: in particular, the protein expression in the presumptive scales of the wing tissue, and the pleiotropic effect of the knock-outs on both eyes (a function conserved with *Drosophila*) and in the wings.

***BarH-1* appear to have evolved as a repressor of pterin granule formation in the wings, and a transposon may drive this gain of function, which is very interesting.**

Overall I find the article to be a great evolutionary biology story that combines

explanation about the why and how of the Alba trait. I think this is exactly the kind of discovery that could go into evolutionary biology textbooks: not only it put in the spotlight an example of a pleiotropic trait, possibly under balance selection to maintain its polymorphism, it is also a neat example of a transposon-mediated novelty.

I especially like the key experimental workflow that goes from phenotypes to genotypes (forward genetics): from mapping to a possible gene, to a characterization of that gene. It is similar to another recent article (Westerman Current Biology 2018) which identified another transcription factor switching yellow/white color in completely different butterflies bearing another kind of pigments. The two stories will be interesting to compare in the classroom for student interested in evolutionary genetics and functional genomics, so I plan to personally use these two stories for pedagogic goals.

It will be interesting to determine if the transposon is found in other species thought to display this ancient Alba polymorphism, and also, perhaps to knock-out the transposon itself, but this is well beyond the scope of the present study, which I recommend for publication.

We thank the reviewer for their kind words. We are glad they will find the work useful in their classroom. We agree the evolutionary history of the locus and knocking out the transposon is indeed of interest for future investigations.

Best Regards,

Alyssa Woronik and Chris Wheat on behalf of the co-authors